# High-throughput ab initio design of atomic interfaces using InterMatch

Eli Gerber [1] ✉, Steven B. Torrisi[2,3], Sara Shabani [4], Eric Seewald[4], Jordan Pack[4], Jennifer E. Hoffman [2,5], Cory R. Dean [4], Abhay N. Pasupathy [4] & Eun-Ah Kim [6]

Forming a hetero-interface is a materials-design strategy that can access an astronomically large phase space. However, the immense phase space necessitates a high-throughput approach for an optimal interface design. Here we introduce a high-throughput computational framework, InterMatch, for efficiently predicting charge transfer, strain, and superlattice structure of an interface by leveraging the databases of individual bulk materials. Specifically, the algorithm reads in the lattice vectors, density of states, and the stiffness tensors for each material in their isolated form from the Materials Project. From these bulk properties, InterMatch estimates the interfacial properties. We benchmark InterMatch predictions for the charge transfer against experimental measurements and supercell density-functional theory calculations. We then use InterMatch to predict promising interface candidates for doping transition metal dichalcogenide $MoSe_2$. Finally, we explain experimental observation of factor of 10 variation in the supercell periodicity within a few microns in graphene/$\alpha$-$RuCl_3$ by exploring low energy superlattice structures as a function of twist angle using InterMatch. We anticipate our open-source InterMatch algorithm accelerating and guiding ever-growing interfacial design efforts. Moreover, the interface database resulting from the InterMatch searches presented in this paper can be readily accessed online.

With increasing control in interface fabrication, interfacial systems form an arena of limitless possibilities[1]. Recent developments with moiré heterostructures[2] further enlarged the phase space to include the twist angle. However, the vast space of possibilities also implies it is crucial to go beyond serendipitous discoveries and empirical explorations to effectively harness the intrinsic potential of interfacial systems. The traditional approach to theoretically studying interfaces is to carry out density-functional theory (DFT) calculations on a supercell system consisting of two materials[3–7]. While such approaches are rigorous, computational limitations regularly require imposing unnatural strain to form a periodic structure. Moreover, the $\mathcal{O}(N^3)$ scaling of DFT in the

number of electrons $N$ for each such calculation prohibits a comprehensive exploration. Some of us recently proposed an intermediate scale approach called Mismatched INterface Theory (MINT)[8], which can predict charge transfer and natural strain approximating one layer of the interface using finite-size scaling of atomic clusters. While MINT calculations are computationally affordable, each calculation typically takes one or more days, long enough time to prevent an exhaustive search. Therefore, a comprehensive and fast approach to scanning the relevant phase space of interfacial combinations is greatly needed.

With advancements in widely available comprehensive materials databases[9–21], it is timely to establish a high-throughput approach to

[1]School of Applied and Engineering Physics, Cornell University, Ithaca, NY 14853, USA. [2]Department of Physics, Harvard University, Cambridge, MA 02138, USA. [3]Energy & Materials Division, Toyota Research Institute, Los Altos, CA 94022, USA. [4]Department of Physics, Columbia University, New York, NY, USA. [5]John A. Paulson School of Engineering and Applied Sciences, Harvard University, Cambridge, MA 02138, USA. [6]Department of Physics, Cornell University, Ithaca, NY 14853, USA. ✉e-mail: eg587@cornell.edu

interface design that can leverage the information contained in these databases to make predictions of interface physics. Indeed, efforts to make interfacial predictions using bulk material databases are beginning to emerge[21–24]. However, so far, the existing approaches aid growth and calculation decisions for a specific pair of materials rather than allowing for a comprehensive query to yield fast, approximate predictions over a wide range of possible interfaces.

In this paper, we introduce InterMatch, which uses information readily available from preexisting materials databases such as the Materials Project and 2DMatPedia databases to predict charge transfer[25], strain[26], stability[27], and optimal superlattice[28] of an atomic interface. Using these predictions, InterMatch can narrow the candidate pool from $C > 10^6$ to ~10 that can then be investigated in greater detail using MINT or supercell DFT (see Fig. 1a). We first illustrate how two branches of InterMatch predict the charge transfer and optimal superlattice after querying the entries of the Materials Project for each of the constituents of the interface. We then benchmark InterMatch predictions for the charge transfer against experimental measurements and supercell DFT predictions. We then employ InterMatch's branches to address two bottleneck problems obstructing design of interfaces towards the goal of discovering new physics: the problem of doping transition metal dichalcogenides and the problem of predicting stable interface structure, applied to the graphene/$\alpha$-RuCl$_3$ system. We comment on many other classes of interfaces that can be optimized using InterMatch. The goal of InterMatch is not to improve the precision of computational methods for predicting these properties, but rather to approach interface design in a data-driven manner by performing exhaustive searches on a previously inaccessible scale, and maximizing existing community contributions to serve a broad user base and accelerate interface design.

## Results

Starting with ab initio materials data, InterMatch performs high-throughput screening of possible heterostructures via pairwise calculation of desired interface properties including charge transfer $\Delta n$,

strain tensor $\bar{\varepsilon}$, optimized superlattice vectors $\mathbf{v}_1$, $\mathbf{v}_2$, and number of atoms $N$. Once InterMatch identifies a promising pool of candidate combinations, one can make a more in-depth analysis of the smaller pool using MINT or supercell DFT (See Fig. 1a). The InterMatch algorithm has two branches to predict two key electronic and mechanical characteristics of candidate interfaces: charge transfer and optimized superlattice structure (See Fig. 1). One branch is devoted to calculating charge transfer (Fig. 1c, d), and the other is devoted to optimizing supercell structure by minimizing the number of atoms and elastic energy (Fig. 1e, f). For the first branch that estimates the direction and magnitude of charge transfer, we use a simple model to describe the Fermi level shifts occurring in each material when they are brought together in proximity[29]. Figure 1c shows how the Fermi level shifts are determined: systems 1 and 2 are designated as donor or acceptor based on their relative Fermi levels $E_F^1$ and $E_F^2$, and the difference of integrals over $g_1(E)$ and $g_2(E)$

$$\int_{E_F'}^{E_F^1} dE \, g_1(E) = \int_{E_F^2}^{E_F'} dE \, g_2(E) \tag{1}$$

is minimized to determine the equilibrium Fermi level $E_F'$. We take interaction between the two systems at the interface into account in the estimation of the charge transfer $\Delta n$ using a simple capacitor model[30]. Specifically, we model the interface as a parallel plate capacitor with the separation $d$ given by the sum of the largest van der Waals radii of the species in each system 1 and 2 (See Fig. 1d). The charge transfer depends on the equilibrium Fermi level $E_F'$ and the distance $d$ as $e\Delta n = \varepsilon_0 E_F'/d$.

The second branch of the InterMatch algorithm sketched in Fig. 1e, f constructs optimal supercells from a pair of queried systems by calculating strain and elastic energy at their interface over a series of supercell configurations. Given the lattice vectors of system 1, $\mathbf{a}_1$, $\mathbf{b}_1$, and those of system 2, $\mathbf{a}_2$, $\mathbf{b}_2$, the algorithm searches for pairs of near-

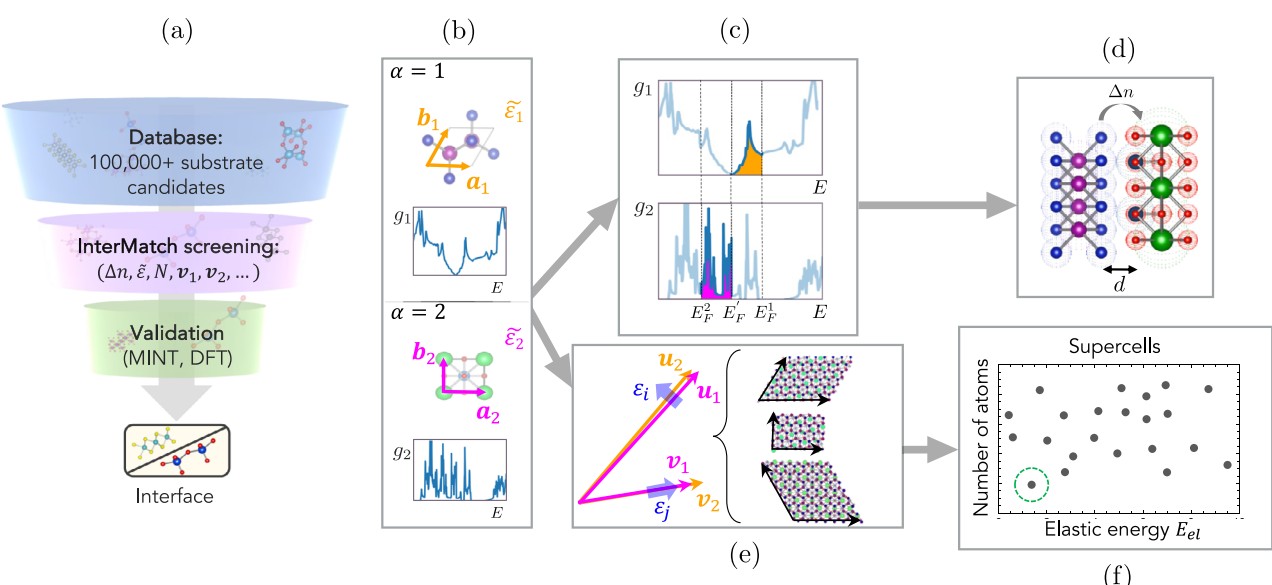

**Fig. 1 | Role of InterMatch and algorithm overview. a** Role of InterMatch in the materials discovery process. After materials' data are queried from major existing materials databases, InterMatch performs pairwise high-throughput screening calculations of interfacial properties, producing output that can be used as input for ab initio verification of the optimized interface candidates. **b** Input from the bulk database. Lattice vectors $\vec{a}_\alpha$, $\vec{b}_\alpha$, density-of-states $g_\alpha$, and elastic tensors $\bar{\varepsilon}_\alpha$ of systems $\alpha = 1, 2$, shown in the top and bottom subpanels, respectively. **c** $E_F^\alpha$ are bulk Fermi levels and $E_F'$ is new equilibrium Fermi level. **d** $\Delta n$ is the transferred charge density and $d$ is the interlayer separation between the two systems, taken to be the sum of the largest van der Waals radii of the species in each system 1 and 2. **e** Superlattice vectors (left) and candidate supercells (right). Superlattice vectors $\mathbf{v}_i$ (orange arrows) and their near-equivalent vectors $\mathbf{u}_i$ (magenta arrows). Candidate supercells, formed in each basis by combining {$(\mathbf{v}_i, \mathbf{u}_i), (\mathbf{v}_j, \mathbf{u}_j)$} pairs specify the strain $\varepsilon_i$ (blue arrows). **f** Optimal supercell minimizes the elastic energy and the number of atoms in the cell. **d, f** These are the outputs of the InterMatch program.

equivalent superlattice vectors $\{(\mathbf{u}_1, \mathbf{v}_1), (\mathbf{u}_2, \mathbf{v}_2)\}$:

$$\mathbf{u}_i = \mathbf{M}^i_{11}\mathbf{a}_i + \mathbf{M}^i_{12}\mathbf{b}_i$$
$$\mathbf{v}_i = \mathbf{M}^i_{21}\mathbf{a}_i + \mathbf{M}^i_{22}\mathbf{b}_i \qquad (2)$$

where $\mathbf{M}^i$ is a $2 \times 2$ matrix of integer coefficients for the system $i$ and the near-equivalence is defined by

$$\mathbf{M}^1 = (\tilde{\varepsilon}^2 + 1)\mathcal{R}_\theta \mathbf{M}^2. \qquad (3)$$

Here $\mathcal{R}_\theta$ is an in-plane rotation matrix by angle $\theta$ and $\tilde{\varepsilon}^2$ is the strain tensor resulting from straining the Bravais lattice of system 2 to match that of system 1.

We choose system 2 to be the material with the smallest elements of the stiffness tensor $\mathbf{C}$ (queried from the Materials Project) in the strain direction. InterMatch then computes the elastic energy $E_{el} = \frac{1}{2}C_{ijkl}\varepsilon_{ij}\varepsilon_{kl}$ for the superlattice candidate according to classical elastic plate theory[26]. The optimal supercell is determined by simultaneously minimizing the elastic energy $E_{el}$ and the number of atoms. By default, we prioritize minimizing the number of atoms in the cells so long as there is no lattice deformation that exceeds 10% of the original lattice constant, but these search parameters are adjustable by the user depending on the goal of the screening.

We note that the user should carefully consider these parameters for optimization depending on the degree of strain/elastic energy one is prepared to accommodate in the final system and verify if larger supercells are not more realistic than a smaller, more distorted lattice.

The use of elastic energy goes beyond previous approaches for finding the superlattice[31,32], which only consider geometric strain. We note that reconstructions and defects are presently beyond the reach of any high-throughput approaches. Hence, one should use InterMatch with caution for properties dependent on long-range interactions.

Now, we demonstrate how elastic energy considerations can make a difference in optimization of the superlattice. Consider the $MoSe_2/ZrTe_3$ interface. The primitive unit cells of $MoSe_2$ and $ZrTe_3$ are shown in Fig. 2a, b, respectively, along with the diagonal components of their stiffness tensors $C_{ii}$ in Voigt notation. The anisotropy of the $ZrTe_3$ stiffness tensor is such that the energetic cost to deforming $ZrTe_3$ along the direction of $C_{22}$ far exceeds the cost of an equivalent deformation along the direction of $C_{11}$. Cells 1 and 2 in Fig. 2c, d result from an InterMatch search for low-area, low-strain $MoSe_2/ZrTe_3$ supercells. The two cells are identical in number of atoms and geometric strain $\varepsilon_{av}$ (shown in Fig. 2e), however, cell 1 is favored energetically due to the different strains required to make each supercell commensurate with $MoSe_2$ (Fig. 2f).

We now benchmark charge transfer predictions by InterMatch against experimentally measured charge transfer in known interfaces. Figure 3a shows a comparison of InterMatch predictions of charge transfer with experimentally obtained values for several interfaces:

$LaAlO_3/SrTiO_3(1\ 1\ 0)$[33], $GR/\alpha$-$RuCl_3$[34], $GR/Pt(111)$[35], $MoS_2/MgAl_2O_4$[36], $MoS_2/TiO_2$[37], and $MoS_2/MoO_3$[37]. The magnitudes of the $\Delta n$ predicted with InterMatch are at the same order of magnitude as the measured values, especially given experimental error bars, with the exception of $GR/\alpha$-$RuCl_3$. However, spin-orbit coupling effects (absent from our calculations) are known to affect the band structure of $\alpha$-$RuCl_3$[38], altering the band alignment with GR and the resulting charge transfer.

Next, we turn to the application of charge transfer prediction to the problem of doping transition metal dichalcogenides (TMDs). TMDs have emerged as an exciting van der Waals material platform at the intersection of semiconductor physics and strong correlation physics[39]. Due to the spin-valley locking Ising spin-orbit coupling, an exotic $p$-wave superconducting state was proposed for hole-doped TMDs[40]. Recent developments in TMD moiré systems have further extended the phase space of possibilities. However, a major bottleneck against testing these proposals is the difficulty of establishing a good contact. Empirically, it has been established that doping the contact area can significantly improve the contact resistance[41]. However, gate-based doping does not scale well. While successful modulation doping using work function difference was established in graphene/$\alpha$-$RuCl_3$ heterostructures[34], it is desirable to perform an exhaustive search of interface possibilities.

We seek 2D substrates for controlling carrier concentration in $MoSe_2$. We screen all entries of the 2DMatPedia database and 3000 entries from the Materials Project for stable 2D materials composed of elements making up the majority of commercially available semiconductors, semimetals, and metals. We use InterMatch to downsample from 10,000 candidate 2D substrates based on the magnitude of the predicted charge transfer $|\Delta n|$ to $MoSe_2$ in the desired range $\geq 10^{13}$ cm$^{-2}$ (Fig. 3b). We then select from these the compounds with maximum $|\Delta n|$, minimal strain, and minimal above-hull energy (Fig. 3c).

Finally, we benchmark InterMatch predictions against supercell DFT calculations using the optimized supercells generated by InterMatch. Figure 3d shows a comparison of InterMatch predictions with the results from DFT for the interfaces in (c) (for computational details, see the Supplemental Material). The magnitude of the $\Delta n$ prediction from the two approaches are within $10^{13}$ cm$^{-2}$. Moreover, both approaches find overall consistent relative magnitude of charge transfer. Given the high-throughput nature of InterMatch, these agreements encourage using InterMatch as the first pass in searches for optimal heterostructures.

As an example of the power of Intermatch to understand superlattice structure, we consider the graphene/$\alpha$-$RuCl_3$ heterostructure (GR/$\alpha$-$RuCl_3$). This system has attracted great interest due to the presence of strong modulation doping[34] and enhancement of $\alpha$-$RuCl_3$'s proximity to the Kitaev spin liquid phase. However, relatively little attention has been paid to the atomic scale structure of the heterostructure and the possible influence on electronic properties. In order

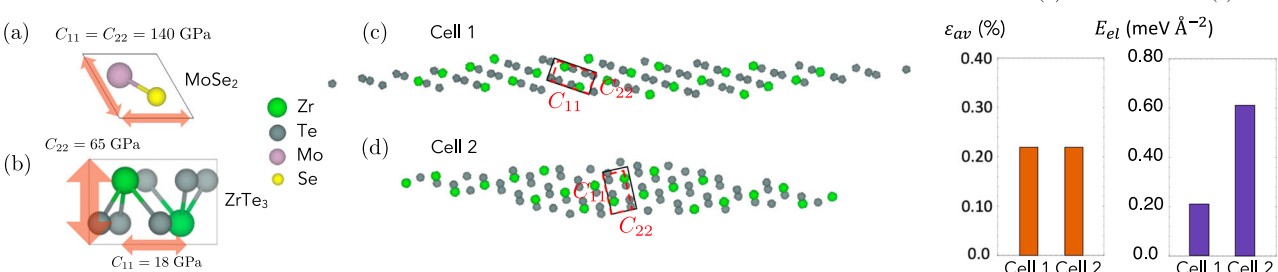

**Fig. 2 | Superlattice structure prediction optimizing elastic energies. a, b** (Top view) Diagonal stiffness tensor components $C_{11}$ and $C_{22}$ of primitive $MoSe_2$ and $ZrTe_3$ unit cells in Voigt notation. **c, d** (Top view) $ZrTe_3$ layer of two candidate $MoSe_2/ZrTe_3$ supercells with the same number of atoms and average strain $\varepsilon_{av}^{ZrTe_3}$. The solid black boxes denote the strained $ZrTe_3$ unit cells and the dashed red boxes are the original unstrained primitive cells. **e** Average strain values $\varepsilon_{av}$ of the $MoSe_2/ZrTe_3$ interfaces in Cells 1 and 2. **f** Elastic energies $E_{el}$ of the interfaces. Source data are provided as a Source Data file.

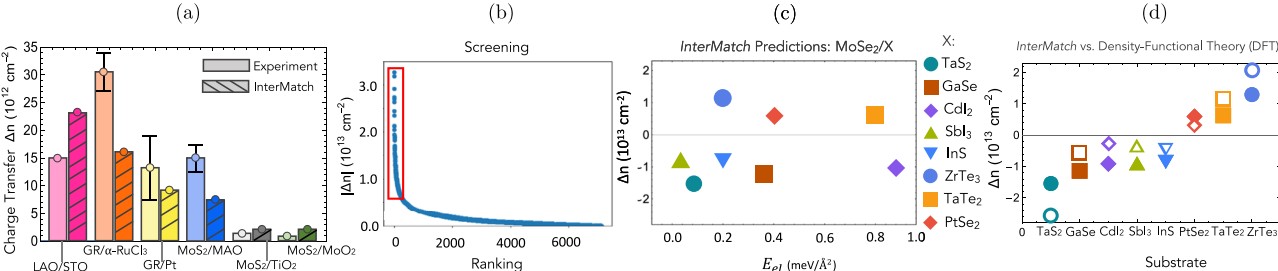

**Fig. 3 | Benchmarking of InterMatch against experiment and density-functional theory. a** Comparison of charge transfer predicted by InterMatch with measured experimental values for interfaces in refs. 33–37. The error bars represent the highest and lowest values of $\Delta n$ observed in experiment. **b** InterMatch screening of over 10,000 2D materials in heterostructure with monolayer $MoSe_2$, ranked in descending order of charge transfer $|\Delta n|$. **c** Substrate selection based on

InterMatch screening results from red box in (**a**) according to $\Delta n$, elastic energy $E_{el}$, and energy above-hull (i.e., the formation energy with respect to composition) to ensure thermodynamic stability. **d** Comparison of InterMatch predictions for $\Delta n$ (solid symbols) with supercell density-functional theory (DFT) calculations (open symbols). Source data are provided as a Source Data file.

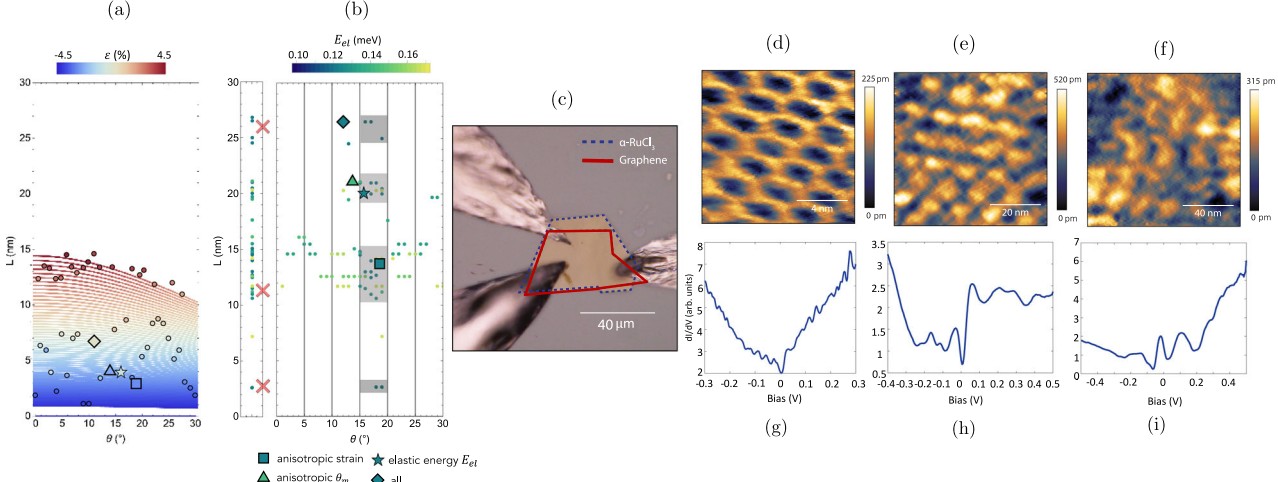

**Fig. 4 | GR/$\alpha$-RuCl$_3$ superlattices: InterMatch predictions vs. experiment.**
**a** Parametric period ($L$) vs angle ($\theta$) curves with isotropic strain $\varepsilon$ in GR according to the model in ref. 42. The color scale indicates the sign and magnitude of isotropic strain in GR for each parametric curve. Overlaid are points denoting InterMatch search results for superlattices having only isotropic strain in GR with $\theta_m = 60°$.
**b** InterMatch predictions for low-energy GR/$\alpha$-RuCl$_3$ superlattice configurations accounting for anisotropic strain, elastic energy $E_{el}$, and $59° \le \theta_m \le 61°$. Left panel is a projection onto the $L$-axis. Shaded gray boxes indicate regions of likely superlattice configurations. Enlarged plot markers (square, triangle, star, diamond) correspond to search criteria that can lead to longer superlattice periods observed

in experiment. Red crosses denote the periodicities extracted from the experiment. Source data are provided as a Source Data file. **c** Optical image of the measured device of GR/$\alpha$-RuCl$_3$ contacted with bismuth indium tin for the STM measurements. The blue and red dashed lines show the boundary of $\alpha$-RuCl$_3$ and graphene, respectively. **d**–**f** STM topographic images (in pm) of GR/$\alpha$-RuCl$_3$ on 2.7 nm (set points of −100 mV and −100 pA), 11.7 nm and 25.7 nm (set points of −1 V and −50 pA) moiré patterns due to atomic reconstruction. **g**–**i** dI/dV measurements corresponding to the three moiré patterns in (**d**)–(**f**) showing strong resonances dependent on moire wavelengths.

to study this experimentally, we used scanning tunneling microscopy (STM) to investigate the properties of GR/$\alpha$-RuCl$_3$ heterostructures created by mechanical exfoliation and colamination, as shown in Fig. 4c. The angle between the $\alpha$-RuCl$_3$ substrate and graphene was not intentionally controlled. Shown in Fig. 4d, f are a set of STM topographs taken at various locations of the GR/$\alpha$-RuCl$_3$ heterostructure. The locations are within a few microns of each other on the sample shown in Fig. 4c. Intriguingly, all three of the regions show moiré patterns with large wavelengths−2.7 nm in Fig. 4d, 11.7 nm in Fig. 4e and 25.7 nm in Fig. 4f. All three of these wavelengths are much larger than the wavelength set by the difference in lattice constants. Using InterMatch, we perform a comprehensive mapping of the space of superlattice configurations spanned by ($\theta, L, E_{el}$) where $\theta$ is the twist angle, $L$ is the moiré period, and $E_{el}$ is the elastic energy of the interface. The resulting spectrum of low-energy superlattice configurations for $0° \le \theta \le 30°$ and $0$ nm $\le L \le 30$ nm is shown in Fig. 4b. For reference, we compute the series of parametric period/angle curves of GR/$\alpha$-RuCl$_3$ moirés using the model from ref. 42, in which only GR is allowed to be

strained, and only isotropic compression/expansion of the GR layer is permitted. These curves are shown in Fig. 4a. We then perform an InterMatch superlattice search with the same constraints, allowing only isotropic strain in the GR layer. These superlattices are represented by points overlaid on the parametric curves in Fig. 4a. Under these constraints, both calculations predict moiré length scales only on the order of the shortest ones observed in the experiment to be stable. We proceed to refine the model by introducing (i) small (<1%[43]) anisotropic strains in either layer (ii) elastic energy as a stability criterion and (iii) small (±1°) deviations in the moiré angle $\theta_m$ from 60°. The lattices predicted under these constraints are the ones shown in Fig. 4b. The elastic energy scale was chosen to reflect typical strain energies observed in GR moiré bilayers[43,44]. While the results shown in both Fig. 4a, b predict some superlattices with relatively low (<1%) strain, introducing properties (i)−(iii) shows a broader spectrum of length scales to be favorable (for further details and examples of how different constraints affect superlattice predictions, see SM Section V). In particular, we find additional length scales that minimize the

interfacial elastic energy and occur within a 5° range between 15° and 20° (gray boxes in Fig. 4b). Three of the four length scales coincide with those observed in STM at $L = 2.7, 11.7, 25.7$ nm, shown in Fig. 4d–f. Furthermore, the InterMatch search algorithm does not rely on solving diophantine equations, making it more efficient than conventional methods while simultaneously allowing one to account for realistic physical properties of the interfacial structure.

Correctly identifying energetically favorable GR/$\alpha$-RuCl$_3$ super-lattices over a narrow range of twist angles showcases InterMatch's capability to predict interfacial structure of complex (e.g., extremely lattice- and elastically-mismatched) systems. We note that in this example, the interface consists of 2D materials, and that there are several risks associated with severing different bonds to produce a particular bulk termination in a 3D system (such as large surface reconstruction and dangling bonds, for example). These can have dramatic effects on the system's electronic and geometric structure, requiring more detailed calculations. Due to these effects, we do not recommend using InterMatch to predict bulk-bulk facets.

The presence of an atomic reconstruction at the interface of GR/$\alpha$-RuCl$_3$ can have dramatic consequences for the spectroscopic properties of the material. Shown in Fig. 4g–i are scanning tunneling spectra averaged over the regions shown in Fig. 4d–f. These spectra show dramatic differences from the simple expectation for a doped Dirac spectrum, as might be expected from charge transfer alone. Instead, we observe strong resonances in all three regions, with the spacing between resonances following the expectation from Landau levels on a Dirac spectrum. Previously, such spectra have been observed when graphene has a periodic buckling[45], where it was ascribed to periodic strain in the material. In our case, apart from the strain associated with the moiré lattice[46], we expect that there will also be strong periodic variations in the doping[47] that contribute to the formation of resonances. In summary, we introduce and demonstrate InterMatch, a high-throughput computational framework and database for predicting charge transfer, strain, and superlattice of an interface between two arbitrary materials. Charge transfer allows heterostructure-based modulation doping[34], which can guide device fabrication and contact design[41]. Efficiently determining the smallest energetically favorable commensurate supercells from a wide variety of interface configurations is crucial for accelerating ab initio studies. We showcase the use of InterMatch by identifying high-charge transfer substrates for doping TMDs, and by predicting equilibrium moiré superlattice configurations for the lattice-mismatched GR/$\alpha$-RuCl$_3$ interface that are validated by STM measurements. The presence of such long-wavelength super-lattice modulations at van der Waals interfaces present new opportunities to tailor bandstructure using materials that do not have a close match in lattice constants. The evolving interface database provides open access to InterMatch results, which we hope will help guide future exploration of interfacial systems.

To broadly benefit the community, we made the InterMatch code openly accessible at https://doi.org/10.5281/zenodo.6823973[48]. Moreover, we tabulate InterMatch results in an open-access "interface database" directly integrated with the Materials Project via the MPContribs platform. At the time of writing, the database contains ~200,000 interfaces (and counting) in simple JavaScript Object Notation that are queryable and sortable according to the chemical composition of either constituent system, charge transfer, strain, and optimized supercell size. In addition, we generate crystallographic information files of interface supercells with InterMatch, which may be readily accessed from the database and used as inputs for DFT or other first principles studies.

## Methods

### InterMatch code
The InterMatch code is written in Python 3.7 and makes extensive use of pymatgen[49], an open-source Python package of the Materials Project, for the manipulation and analysis of various structures of interest. The code is continuously being developed, and the latest version can be obtained at https://doi.org/10.5281/zenodo.6823973[48]. We aim to provide an efficient scheme for computing interface properties capable of screening a significant fraction of combinations of existing Materials Project structure entries, returning the results in real-time (typical run time for the calculation of a single interface is $10 \pm 5$ s on a 2011 Macbook Air (model number MC965LL/A) with an Intel Core i5 processor (I5-2557M) at 1.7 GHz using 4 GB of RAM, running macOS Sierra 10.12.6).

### Computational details
All ab initio DFT calculations were carried out within the total-energy plane wave density-functional pseudopotential approach, using Perdew–Burke–Ernzerhof generalized gradient approximation functionals[50] and optimized norm-conserving Vanderbilt pseudopotentials in the SG15 family[51]. Plane-wave basis sets with energy cutoffs of 30 Hartree were used to expand the electronic wave functions. We used fully periodic boundary conditions and a $8 \times 8 \times 1 k$-point mesh to sample the Brillouin zone. Electronic minimizations were carried out using the analytically continued functional approach starting with a LCAO initial guess within the DFT + + formalism[52], as implemented in the open-source code JDFTx[53] using direct minimization via the conjugate gradients algorithm[54]. All unit cells were constructed to be inversion symmetric about $z = 0$ with a distance of ~60 bohr between periodic images of the MoSe$_2$ surface, using coulomb truncation to prevent image interaction.

## Data availability
The authors declare that the data supporting the findings of this study are available within the paper, its Supplementary Information file, and at MPContribs[55]. All other InterMatch data generated in this study are presented in the figures of the main text, and Supplementary Information are provided in the Source Data file. Source data are provided with this paper.

## Code availability
The authors declare that the code used to support the findings of this study is available and openly accessible under the MIT License at https://doi.org/10.5281/zenodo.6823973[48].

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

## Acknowledgements

The authors thank Kin Fai Mak, Jie Shan, Stephen Carr, Patrick Huck, Matthew Horton, Jason Munro, and Vidya Madhavan for helpful discussions. EAK and JH were supported by MURI grant FA9550-21-1-0429. E.G. was supported by the Cornell Center for Materials Research with funding from the NSF MRSEC program (DMR-1719875). S.B.T. was supported by the Department of Energy Computational Science Graduate Fellowship under grant DE-FG02-97ER25308. The computation was done using

the high-powered computing cluster WALLE2 that was established through the support of Gordon and Betty Moore Foundation's EPiQS Initiative, Grant GBMF10436 to E.A.K. and the New Frontier Grant from Cornell University's College of Arts and Sciences and hosted and maintained by Cornell Center for Advanced Computing. STM experiments were supported by NSF DMR-2004691 (A.N.P.) and AFOSR via grant FA9550-21-1-0378 (S.S., E.S.). Sample synthesis for STM measurements was supported by the NSF MRSEC program through Columbia in the Center for Precision-Assembled Quantum Materials (PAQM), grant number DMR-2011738.

## Author contributions

E.G. performed the calculations and created the database. E.G. and S.B.T. designed the algorithm and wrote the code. E.G., S.B.T., A.N.P. and E.-A.K. wrote the paper. S.S., E.S. and A.N.P performed the experiment and analyzed the data. J.P. and C.R.D. performed the sample fabrication. J.E.H. and E-A.K. supervised the project.

## Competing interests

The authors declare no competing interests.
