## [Peer Review File · Nature Communications]

Reviewers' Comments:

Reviewer #1:

Remarks to the Author:

I'm reviewing the article Gerber et al "High-Throughput Ab Initio Design of Atomic Interfaces using InterMatch". While I do share interest in computational discovery, I cannot recommend acceptance in Nature Communications for the following reasons.

- 1) The article is mostly a derivative work and as such does not contain the level of novelty required for the journal
- 2) There are no new ideas, and no novel science is introduced whatsoever, that was not known before. It is an incremental mosaic of pieces already available before.

While I do realize the magnitude of the work, there is not enough novelty. I believe Nature Communications is not the proper venue to publish this work. I suggest transfer to Scientific Reports/npj-Comp. Mater. (there are also theory-experimental articles there), or even a more focused journal such as Comp. Mat. Sci.

Reviewer #2:

Remarks to the Author:

The manuscript is concerned with a high-throughput algorithm (called "InterMatch" by the authors) that is used to find promising heterogeneous material interfaces based on simple predictions of charge transfer and elastic energy due to lattice mismatch. Predictions are based on data from pre-existing reference databases. Some example use cases for the code are shown.

Comments:

* The algorithm optimizes for both minimum number of atoms and minimum elastic energy. The former is not a physical criterion, but a technical one: One probably wants to keep the systems small for potential DFT calculations or similar. Figure 1(f) is deceptive here: it suggests that the minimum elastic energy coincides with the minimum number of atoms, but that is not necessarily true. It is unclear to me how this tradeoff is made and if it is sensible. The authors must elaborate on this.

* The calculation of strain energy does not consider reconstructions/defects that could reduce the strain energy. It is well known, for example, that certain films grown on substrates reduce the mismatch strain by introducing dislocations. This could invalidate the screening significantly. The authors must discuss this limitation: maybe it does not matter for certain material classes of interest? Reconstructions should also affect the electronic states and thus the estimation of the charge transfer, but the relevance of this would depend on the material system, too. The reader would need a way to know if the screening is trustworthy for a specific system of interest (or at all).

* Figure 2(a): The differences between the experiments and InterMatch seem quite large, in fact. Only GR/Pt(111) match within the error bars. The authors say that InterMatch predictions are of the same order of magnitude as the experiments, but then one should also recognize that all the experimental values for the different system are of the same order of magnitude. So "order of magnitude" seems insufficient.

* Verification is performed by comparing DFT with InterMatch predictions for a range of heterostructures. These seem fine, but there is the question of validating the proposed structures. How can we be sure that InterMatch selected a reasonable subset of structures, since there is obviously no calculation of the structures that were discarded? Maybe those would have been better? I understand that this is a necessary limitation of high-throughput screening, but there should at least be some critical discussion of how one would evaluate this.

* The example case of GR/ α -RuCl₃ is not very clear. The highlight is supposed to be that predicted moiré length scales match to experiment, when restricting the twist angle to the

experimental range. The text does not give many details what "identify" in "we identify four prominent moire length scales..." (l. 227ff) means. Is this trivial (i.e., already clear due to geometry) or why is this proof of the good performance of InterMatch?

Minor comments:

At the end of the abstract: the authors mention "MPContribs", but it is unclear what that is. It should probably be removed there and only mentioned in the main text together with an explanation and/or citation/link.

l. 34: What does "real time" mean in this context? I think this is supposed to mean "reasonable time".

l. 54: The big-O notation does not make sense to me, I know this as a notation for scaling, i.e., $O(10^6) = O(10) = O(1)$, in contrast to, e.g., $O(n)$, $O(\log(n))$, $O(n^2)$, etc. I would just remove the "O".

Ref. 24: This is missing bibliographic information, I cannot find this paper.

Supplemental, l. 37: The CPU is identified only as "1.7 GHz Intel Core i5 processor at 1333 MHz". First, the Core i5 name is used since years and can refer to quite different CPU generations. We would need the actual model name to evaluate the actual performance of the CPU (and no, GHz does not mean anything anymore since at least a decade). Second, does it run at 1.7 GHz or 1.333 GHz?

Reviewer #3:

Remarks to the Author:

This paper is very insightful. It aims to solve one of the most challenging problems in theoretical material science.

The authors propose a simple and straightforward approach to predicting interface properties. The method itself doesn't seem to be very complicated, but it looks pretty effective in narrowing a short list of candidates for more computationally extensive approaches (like DFT) to further investigate.

The paper lists a few interface candidates and conducts comprehensive comparisons of the proposed approach against theoretical simulations and experiments. This clearly demonstrates the effectiveness and advantages of the approach. Specifically, the charge transfer predictions are not too far off from experimental measurements even without considering the spin-orbit coupling effects.

Here are a few additional analyses that can help readers to better understand the methods:

1. In addition to comparing charge transfer with DFT results, other thermodynamic properties can also be compared. These include interlayer separation distance, stable interface lattice constant, and more.
2. Extending the benchmark candidates to other materials groups, such as transition metal alloys and transition metal oxides, would be beneficial.
3. It is unclear to me whether this method can discover the optimal combination of facets from two provided bulk structures, and whether the facet prediction from InterMatch agrees with DFT or experimental results.

In general, I find the paper to present an intriguing perspective and would recommend it for acceptance by the editors.

Response to the Referees

Summary of Changes

- We revised Figure 1(f) and the corresponding part of the main text (l.129-133) to clarify the algorithm’s cell selection criteria.
- We modified the main text (l.136, 1.482-485) to include a comment on reconstructions and defects and when they might affect InterMatch predictions.
- We included additional experimentally well-known interfaces with relatively lower charge transfer in our benchmarking of InterMatch charge transfer predictions to Figure 2(a) of the main text.
- To Figure 2(d) of the main text, we added DFT verification of charge transfer for all systems shown in Figure 2(c) of the main text.
- We clarified the GR/ α -RuCl₃ superlattice selection procedure in l.237-271 of the main text, added a comparison of moiré length scale predictions with conventional methods (Figure 4 (e)-(f)), as well as a discussion of how accounting for elastic energy, strain and moiré angle anisotropy lead to the prediction of longer periodicities. Furthermore, we added quantitative examples and discussion illustrating the effects of accounting for each of these physical parameters in our model to section V of the SM.
- For all systems shown in Figure 2(c) of the main text, we added DFT verification of interface lattice constants and interlayer separation distances to section II of the SM.
- We included a discussion of the limitations of InterMatch to predict bulk-bulk facets in l.276, 1.486-494 of the main text.
- We addressed all minor comments made by the referees in the main text and SM.

Report of Referee 1

I’m reviewing the article Gerber et al “High-Throughput Ab Initio Design of Atomic Interfaces using InterMatch”. While I do share interest in computational discovery, I cannot recommend acceptance in Nature Communications for the following reasons.

1. The article is mostly a derivative work and as such does not contain the level of novelty required for the journal.
2. There are no new ideas, and no novel science is introduced whatsoever, that was not known before. It is an incremental mosaic of pieces already available before.

While I do realize the magnitude of the work, there is not enough novelty. I believe Nature Communications is not the proper venue to publish this work. I suggest transfer to Scientific Reports/npj-Comp. Mater. (there are also theory-experimental articles there), or even a more focused journal such as Comp. Mat. Sci.

Response: We respectfully disagree with the referee. The new idea is to take a data-centric approach to designing 2D interfaces by starting from an exhaustive and systematic search with the new tool of InterMatch. The whole point of the work is to make the data curated throughout the Materials Genome era serve a wide user base and rapidly growing frontier of interface design. We believe the paper will have the impact worthy of publication in Nature Communications, based on our interactions with the community and on the other two referee’s assessments.

Report of Referee 2

The manuscript is concerned with a high-throughput algorithm (called “InterMatch” by the authors) that is used to find promising heterogeneous material interfaces based on simple predictions of charge transfer and elastic energy due to lattice mismatch. Predictions are based on data from pre-existing reference databases. Some example use cases for the code are shown.

1. The algorithm optimizes for both minimum number of atoms and minimum elastic energy. The former is not a physical criterion, but a technical one: One probably wants to keep the systems small for potential DFT calculations or similar. Figure 1(f) is deceptive here: it suggests that the minimum elastic energy coincides with the minimum number of atoms, but that is not necessarily true. It is unclear to me how this tradeoff is made and if it is sensible. The authors must elaborate on this.

Response: We recognize the original manuscript was unclear regarding the explanation of Fig 1(f) and the corresponding part of the algorithm. In practice, the code allows the user to set independent tolerance for the maximum number of atoms and the maximum elastic energy or strain allowed in a supercell search. By default, we prioritize minimizing the number of atoms in the cells so long as there is no lattice deformation that exceeds 10% of the original lattice constant. Under the default settings, the algorithm produces a list of configurations that minimize first and foremost the number of atoms, ranked in ascending order of strain energy. The distribution of points in the previous Fig 1(f) was indeed unclear in suggesting some correlation between the two variables, as pointed out by the referee. Prompted by the referee’s comments we have revised the figure to better reflect the cell selection procedure, and clarified the above point in 1.129-133 the main text.

2. The calculation of strain energy does not consider reconstructions/defects that could reduce the strain energy. It is well known, for example, that certain films grown on substrates reduce the mismatch strain by introducing dislocations. This could invalidate the screening significantly. The authors must discuss this limitation: maybe it does not matter for certain material classes of interest? Reconstructions should also affect the electronic states and thus the estimation of the charge transfer, but the relevance of this would depend on the material system, too. The reader would need a way to know if the screening is trustworthy for a specific system of interest (or at all).

Response: We agree with the referee that indeed not being able to include reconstructions and defects is a shortcoming. As the referee must know, reconstructions and defects are problems beyond the reach of high-throughput approaches.

Prompted by the above comment, we clarified (1.136, 1.482-485) that users must be aware that InterMatch predictions should be taken with caution for the properties involving long-range interactions as a result. We anticipate InterMatch predictions to be reliable for properties concerning localized interactions (e.g. a transition associated with one orbital, or a particular catalytically active site).

3. Figure 2(a): The differences between the experiments and InterMatch seem quite large, in fact. Only GR/Pt(111) match within the error bars. The authors say that InterMatch predictions are of the same order of magnitude as the experiments, but then one should also recognize that all the experimental values for the different system are of the same order of magnitude. So “order of magnitude” seems insufficient.

Response: In general, the InterMatch aims to provide a first pass to narrow the pool of candidate systems. The InterMatch aims to predict trends correctly and provide a correct order of magnitude estimate. Once InterMatch narrows the pool, more detailed and rigorous quantitative calculations should be carried out, as we indicate in Figure 1(a). Prompted by the referee’s remark we clarified the goal of InterMatch in 1.70-76 of the main text. Furthermore, we have included several experimentally well-known interfaces having over an order of magnitude lower charge transfer such as $\text{MoS}_2/\text{MoO}_3$ and $\text{MoS}_2/\text{TiO}_2$ in Figure 2(a) to demonstrate that InterMatch predictions are consistent with a broader variety of systems with different band alignments (i.e not exclusively “high” charge transfer interfaces).

4. Verification is performed by comparing DFT with InterMatch predictions for a range of heterostructures. These seem fine, but there is the question of validating the proposed structures. How can we be sure that InterMatch selected a reasonable subset of structures, since there is obviously no calculation of the structures that were discarded? Maybe those would have been better? I understand that this is a necessary limitation of high-throughput screening, but there should at least be some critical discussion of how one would evaluate this.

Response: We thank the referee for the suggestion. Prompted by the suggestion, we have added DFT verification for the rest of the systems in Figure 2(c). We now fully disclose all the cases’ verification and demonstrate that

to the best of our knowledge all DFT verification of charge transfer, lattice constant, and separation distance agrees with InterMatch predictions to a consistent level of accuracy in Figure 2(d) of the main text and section II of the SM.

5. The example case of GR/ α -RuCl₃ is not very clear. The highlight is supposed to be that predicted moiré length scales match to experiment, when restricting the twist angle to the experimental range. The text does not give many details what “identify” in “we identify four prominent moiré length scales...” (l. 227ff) means. Is this trivial (i.e., already clear due to geometry) or why is this proof of the good performance of InterMatch?

Response: Indeed, the originally submitted text was sparse on the detail. In Fig. 4 (f), the configurations (represented by shaded dots in the figure) are “identified” insofar as they minimize the interfacial elastic energy. Conventional methods only optimize over the geometry (i.e. the difference in reciprocal lattice vectors of the constituent lattices at a given twist angle) and the isotropic strain. InterMatch is designed to be more flexible by allowing for anisotropy in strain and in moiré angle and it also optimizes over elastic energy.

Prompted by the referee’s comments, we clarified the GR/ α -RuCl₃ superlattice selection procedure in 1.237-271 of the main text, added new calculations comparing moiré length scale predictions with conventional methods from existing literature (Figure 4 (e)-(f)), as well as a discussion of how accounting for elastic energy, strain and moiré angle anisotropy lead to the prediction of longer periodicities. Furthermore, we added quantitative examples and discussion illustrating the effects of accounting for each of these physical parameters in our model to section V of the SM.

Minor comments:

1. At the end of the abstract: the authors mention “MPContribs”, but it is unclear what that is. It should probably be removed there and only mentioned in the main text together with an explanation and/or citation/link.
2. l. 34: What does “real time” mean in this context? I think this is supposed to mean “reasonable time”.
3. l. 54: The big-O notation does not make sense to me, I know this as a notation for scaling, i.e., $O(10^6) = O(10) = O(1)$, in contrast to, e.g., $O(n)$, $O(\log(n))$, $O(n^2)$, etc. I would just remove the “O”.
4. Ref. 24: This is missing bibliographic information, I cannot find this paper.
5. Supplemental, l. 37: The CPU is identified only as “1.7 GHz Intel Core i5 processor at 1333 MHz”. First, the Core i5 name is used since years and can refer to quite different CPU generations. We would need the actual model name to evaluate the actual performance of the CPU (and no, GHz does not mean anything anymore since at least a decade). Second, does it run at 1.7 GHz or 1.333 Ghz?

Response: We thank the referee for this careful reading, and have addressed all minor comments noted above.

Report of Referee 3

This paper is very insightful. It aims to solve one of the most challenging problems in theoretical material science.

The authors propose a simple and straightforward approach to predicting interface properties. The method itself doesn’t seem to be very complicated, but it looks pretty effective in narrowing a short list of candidates for more computationally extensive approaches (like DFT) to further investigate.

The paper lists a few interface candidates and conducts comprehensive comparisons of the proposed approach against theoretical simulations and experiments. This clearly demonstrates the effectiveness and advantages of the approach. Specifically, the charge transfer predictions are not too far off from experimental measurements even without considering the spin-orbit coupling effects.

We thank the reviewer for these positive comments highlighting the impact of our approach and the recommendation for publication. We are also grateful for the referee’s thoughtful questions and suggestions, which have helped us provide a clearer presentation of the manuscript.

Here are a few additional analyses that can help readers to better understand the methods:

1. In addition to comparing charge transfer with DFT results, other thermodynamic properties can also be compared. These include interlayer separation distance, stable interface lattice constant, and more.

Response: We thank the referee for this suggestion. Prompted by the suggestion we have included a comparison of interface lattice constants as well as interlayer separation distances in our DFT verification. We include these new results in section II of the SM.

2. Extending the benchmark candidates to other materials groups, such as transition metal alloys and transition metal oxides, would be beneficial.

Response: We thank the referee for this suggestion. Prompted by the referee's remark we have included some well-known MoSe₂/TMO interfaces in the benchmarking Figure 2(a) of the main text.

3. It is unclear to me whether this method can discover the optimal combination of facets from two provided bulk structures, and whether the facet prediction from InterMatch agrees with DFT or experimental results.

Response: We thank the referee for this comment. There are several risks associated with severing different bonds to produce a particular bulk termination (such as large surface reconstruction and dangling bonds, for example) that can have dramatic effects on the system's electronic and geometric structure, requiring more detailed calculations. Due to these effects, we do not recommend using InterMatch to predict bulk-bulk facets and we have included this disclaimer as a footnote in 1.276, 1.486-494 of the main text.

In general, I find the paper to present an intriguing perspective and would recommend it for acceptance by the editors.

Reviewers' Comments:

Reviewer #2:

Remarks to the Author:

The authors have satisfactorily addressed most of my and the other reviewers' concerns in their response. I have only some minor comments:

* I am still not happy about prioritizing the minimization of the number of atoms in the supercell over other values. In reality, there must be a tradeoff. Still, the verifications presented in the manuscript seem to indicate that the authors' strategy is valid nonetheless in these cases. In order to be most helpful to the readers of the article and users of the software, I would request that the authors add a short comment that the user must carefully consider the parameters for optimization and verify if larger supercells are not more realistic. For example, the 10% strain limit is quite high! It might work, but I want to avoid people blindly accepting these defaults.

* Fig. 2 and Fig. 3 should be switched to match the order they are referenced in the text.

Regarding the comment of Reviewer 1: It is true that the manuscript focuses mostly on method development. However, the example of how InterMatch can be applied to GR/ α -RuCl₃ interfaces is novel and interesting on its own, for example, especially in the expanded form of the updated manuscript. The verification that the simplified methods used by InterMatch actually yield decent results is also of interest to the community. The paper definitely deserves to be published, but I will not further comment on the suitability for Nat. Commun. beyond summarizing the type of content of the paper as done above, since this is ultimately an editorial decision.

Reviewer #3:

Remarks to the Author:

The revisions made by the authors have not altered my initial assessment of this paper. I continue to believe that it presents a valuable data-driven approach that could be beneficial to this field.

The incorporation of additional results by the authors, in accordance with my previous feedback, further supports the effectiveness of this approach. The gaps in lattice constants and interlayer separation distances between InterMatch and DFT persist within an acceptable margin.

Response to the Referees

Summary of Changes

- We added a comment in l.164-168 of the main text cautioning that the user must carefully consider the supercell optimization parameters and verify if larger supercells are not more realistic than allowing higher strains.
- We switched the order of figures 2 and 3 so they appear in the order in which they are referenced in the text.

Report of Referee 2

The authors have satisfactorily addressed most of my and the other reviewers' concerns in their response. I have only some minor comments:

1. I am still not happy about prioritizing the minimization of the number of atoms in the supercell over other values. In reality, there must be a tradeoff. Still, the verifications presented in the manuscript seem to indicate that the authors' strategy is valid nonetheless in these cases. In order to be most helpful to the readers of the article and users of the software, I would request that the authors add a short comment that the user must carefully consider the parameters for optimization and verify if larger supercells are not more realistic. For example, the 10% strain limit is quite high! It might work, but I want to avoid people blindly accepting these defaults.

Response: We thank the reviewer for their insightful comment. In response to their request, we have added a comment in l.164-168 of the main text cautioning that the user must carefully consider the supercell optimization parameters and verify if larger supercells are not more realistic than allowing higher strains.

2. Fig. 2 and Fig. 3 should be switched to match the order they are referenced in the text.

Response: In response to the reviewer's comment we switched the order of figures 2 and 3 so they appear in the order in which they are referenced in the text.

Regarding the comment of Reviewer 1: It is true that the manuscript focuses mostly on method development. However, the example of how InterMatch can be applied to GR/ α -RuCl₃ interfaces is novel and interesting on its own, for example, especially in the expanded form of the updated manuscript. The verification that the simplified methods used by InterMatch actually yield decent results is also of interest to the community. The paper definitely deserves to be published, but I will not further comment on the suitability for Nat. Commun. beyond summarizing the type of content of the paper as done above, since this is ultimately an editorial decision.

Report of Referee 3

The revisions made by the authors have not altered my initial assessment of this paper. I continue to believe that it presents a valuable data-driven approach that could be beneficial to this field.

The incorporation of additional results by the authors, in accordance with my previous feedback, further supports the effectiveness of this approach. The gaps in lattice constants and interlayer separation distances between InterMatch and DFT persist within an acceptable margin.

Response: We thank the reviewer for their feedback and careful reading of the manuscript.